# CipherPrune: Efficient and Scalable Private Transformer Inference

Yancheng Zhang[1], Jiaqi Xue[1], Mengxin Zheng[1]
Mimi Xie[2], Mingzhe Zhang[3], Lei Jiang[4], Qian Lou[1*]

[1]*University of Central Florida*     [2]*University of Texas at San Antonio*
[3]*Ant Research*     [4]*Indiana University Bloomington*

## Abstract

Private Transformer inference using cryptographic protocols offers promising solutions for privacy-preserving machine learning; however, it still faces significant runtime overhead (efficiency issues) and challenges in handling long-token inputs (scalability issues). We observe that the Transformer's operational complexity scales quadratically with the number of input tokens, making it essential to reduce the input token length. Notably, each token varies in importance, and many inputs contain redundant tokens. Additionally, prior private inference methods that rely on high-degree polynomial approximations for non-linear activations are computationally expensive. Therefore, reducing the polynomial degree for less important tokens can significantly accelerate private inference. Building on these observations, we propose *CipherPrune*, an efficient and scalable private inference framework that includes a secure encrypted token pruning protocol, a polynomial reduction protocol, and corresponding Transformer network optimizations. At the protocol level, encrypted token pruning adaptively removes unimportant tokens from encrypted inputs in a progressive, layer-wise manner. Additionally, encrypted polynomial reduction assigns lower-degree polynomials to less important tokens after pruning, enhancing efficiency without decryption. At the network level, we introduce protocol-aware network optimization via a gradient-based search to maximize pruning thresholds and polynomial reduction conditions while maintaining the desired accuracy. Our experiments demonstrate that CipherPrune reduces the execution overhead of private Transformer inference by approximately $6.1\times$ for 128-token inputs and $10.6\times$ for 512-token inputs, compared to previous methods, with only a marginal drop in accuracy. The code is publicly available at https://github.com/UCF-Lou-Lab-PET/cipher-prune-inference.

## 1 Introduction

Transformers (Vaswani et al., 2017) have become the predominant approach for tackling a wide range of machine learning tasks, spanning Natural Language Processing (NLP) (Xue et al., 2024) and Computer Vision (CV) (Zheng et al., 2022) domains. Notably, Transformer-as-a-Service (TaaS) (Radford et al., 2018) has emerged as an effective means for average users to harness the capabilities of sophisticated and accurate Transformers deployed on cloud servers. Privacy has become a major concern, driving a growing demand for privacy-preserving TaaS solutions (Zheng et al., 2023; Hao et al., 2022; Zeng et al., 2023; Santriaji et al., 2025; Lou et al., 2021).

Homomorphic Encryption (HE) (Gentry, 2009) is a promising secure computing technology that protects data privacy by enabling computations on encrypted data without decryption. However, applying HE continuously for deep computation tasks often results in prohibitively high latency (Lou et al., 2019; Lou & Jiang, 2019; Zheng et al., 2024; Zhang et al., 2023a; Jiang et al., 2022). To address this, hybrid HE/Multi-party Computation (MPC)-based techniques (Chen et al., 2022; Zheng et al., 2023; Hao et al., 2022; Zeng et al., 2023; Zhang et al., 2023b; Lu et al., 2025; Pang et al., 2024; Xu et al., 2024b) have been widely adopted for private Transformer inference, as illustrated in Figure 1

---

*Corresponding Author. Email: qian.lou@ucf.edu.

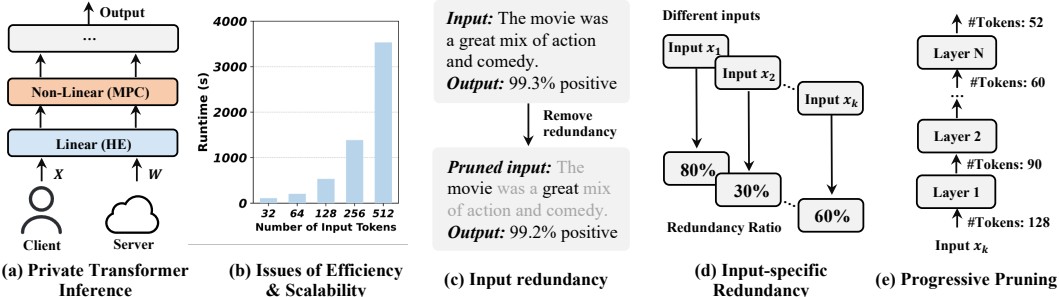

Figure 1: (a) An illustration of HE/MPC-based private inference. (b) The high-latency and scalable challenge of private Transformer models over lengthy inputs. (c) An example of redundant input in sentiment analysis tasks. (d) Demonstration of varying levels of redundancy across different inputs. (e) An example showcasing progressive redundancy pruning.

(a). This hybrid approach achieves state-of-the-art performance by using HE for linear operations and MPC for non-linear operations.

Unfortunately, prior private Transformer inferences (Lu et al., 2025; Pang et al., 2024; Zheng et al., 2023) still suffer from significant latency and poor scalability over long-token inputs. As Figure 1 (b) shows, the prior private inference process for a GPT2 Transformer (Pang et al., 2024) with 128 input tokens extends to $\sim$ 10 minutes. It necessitates the exchange of over 60 gigabytes of data between the server and the client. Furthermore, as the token length increases, the runtime overhead grows super-linearly, indicating poor scalability. This is primarily because the operational complexity of Transformers (Vaswani et al., 2017; Kim et al., 2022) scales quadratically with the number of input tokens. Reducing the number of input tokens without compromising accuracy is essential.

We observe that most inputs contain redundant words/tokens, with varying levels of redundancy across different inputs. As illustrated in Figure 1 (c), in a sentiment analysis task, an input that retains only the tokens *movie* and *great* while removing almost all others still maintains inference confidence and accuracy. We refer to such tokens that can be removed without significantly impacting accuracy as *redundancy*. Meanwhile, the different inputs have various levels of redundancy (Wang et al., 2021; Kim et al., 2022; Yudha et al., 2024). Figure 1 (d) illustrates that some inputs exhibit greater redundancy, while others have less, with this variation being particularly evident across different tasks. Classification tasks typically have more redundancy compared to sequence-to-sequence tasks (Fu et al., 2024). To effectively prune more tokens from longer inputs and potentially reduce the Transformer's quadratic complexity to linear, pruning should be done progressively—that is, tokens should be pruned layer by layer over multiple stages, as illustrated in Figure 1 (e), rather than performing a one-time pruning at the first layer (Wang et al., 2021; Kim et al., 2022; Xu et al., 2024a). Another key observation is that previous private Transformer inference methods, whether relying on precise non-linear activations (Hao et al., 2022; Yudha et al., 2024) or using large-degree polynomial approximations (Lu et al., 2025; Pang et al., 2024) for these activations, continue to suffer from significant execution overhead for non-linear operations. Therefore, replacing non-linear activations or high-degree polynomials with lower-degree polynomials can be beneficial. As shown in Figure 2 , a degree-$d$ polynomial activation for tokens (Figure 2 (a)) can be reduced to a degree-$d_i$ polynomial (Figure 2 (b)), where $d_i \leq d$.

Adopting existing plaintext-level pruning techniques (*PlainPrune*) (Wang et al., 2021; Kim et al., 2022) to accelerate private Transformer inferences presents a formidable challenge. The primary reason is that the tokens are encrypted, and we need to calculate token importance scores layer by layer for specific encrypted inputs. This requires redesigning a new encrypted token pruning protocol. Meanwhile, the polynomial reduction in the encrypted domain poses a similar challenge, and we need to design an encrypted polynomial reduction protocol for efficient private activation. Also, at the network level, we need to learn the pruning and polynomial reduction thresholds while maximizing *efficiency*, ensuring *privacy*, and maintaining the desired level of *accuracy*.

To address these challenges, we introduce CipherPrune, a scalable and efficient framework for private inference that incorporates a secure encrypted token pruning protocol, a polynomial reduction protocol, and tailored Transformer network optimizations. At the protocol level, CipherPrune adaptively prunes unimportant tokens from encrypted inputs in a progressive, layer-by-layer manner. It also applies encrypted polynomial reduction

by assigning lower-degree polynomials to less important tokens post-pruning, thereby improving efficiency without requiring decryption. At the network level, we implement protocol-aware optimization using a gradient-based search, aiming to maximize pruning thresholds and polynomial reduction conditions while preserving the required accuracy. Our experiments show that CipherPrune reduces the execution overhead of private Transformer inference by about $6.1\times$ for 128-token inputs and $10.6\times$ for 512-token

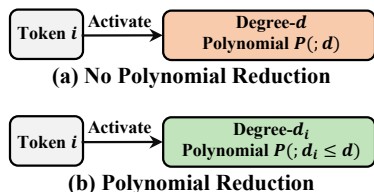

(a) No Polynomial Reduction

(b) Polynomial Reduction

Figure 2: Polynomial Reduction.

inputs compared to prior method (Pang et al., 2024), with only a marginal drop in accuracy.

## 2 PRELIMINARIES

**Token Pruning in Transformers**. To reduce the computational overhead of plaintext Transformers, a range of strategies including efficient architecture design, knowledge distillation, quantization, and both model and token pruning have been developed. Among these, token pruning (Goyal et al., 2020; Kim & Cho, 2020; Wang et al., 2021) stands out for its ability to dynamically reduce token complexity, enhancing efficiency for scalable input lengths. Online token pruning progressively eliminates nonessential input tokens during inference, with recent techniques like learning-based token pruning assigning tunable thresholds to each Transformer layer. These thresholds are fine-tuned during training to maximize the removal of tokens from the sequence.

Given the attention map $Att = SoftMax(QK^T/\sqrt{d}) \in \mathbb{R}^{n \times n}$, $QK^T$ computes the dot product between the query and key vectors, resulting in an $n \times n$ matrix. Dividing by $\sqrt{d}$ scales down the values to avoid excessively large dot products, especially when the dimensionality d is large. The importance score $S \in \mathbb{R}^n$ of $n$ tokens $x_i \in \mathbb{R}^D$ in the input sequence can be computed as: $S[i] = \frac{1}{H}\frac{1}{n}\sum_{h=0}^{H-1}\sum_{j=0}^{n-1} Att^h[j,i]$ (1) where $H$ is the number of attention heads and $Att^h$ is the attention map in the $h$-th head. Importance score $S$ is essentially computed by accumulating attention scores vertically, which indicates the importance of a token across all heads in one Transformer layer.

**Cryptographic Primitives**. Our protocol uses multiple cryptographic primitives including Additive Secret Sharing (ASS), Homomorphic Encryption (HE), and Oblivious Transfer (OT). We reuse partial existing protocols detailed in Appendix B.

- **ASS**. We employ a 2-out-of-2 ASS scheme (Cramer et al., 2015) operating over the ring $\mathbb{Z}_\ell$, where $\ell$ is the bitwidth of the input $x$. ASS partitions $x$ into two distinct random shares $\langle x \rangle_0, \langle x \rangle_1$, where $x = \langle x \rangle_0 + \langle x \rangle_1 \mod \mathbb{Z}_\ell$. The parties, $P_0$ and $P_1$, respectively hold $\langle x \rangle_0, \langle x \rangle_1$. Importantly, it is guaranteed that neither $P_0$ nor $P_1$ can discern the actual value of $x$ (Cramer et al., 2015). ASS lends itself to linear operations, i.e., addition and constant multiplication, without communications.
- **HE**. We leverage the BFV scheme (Brakerski, 2012; Fan & Vercauteren, 2012; Deng et al., 2024), a leveled HE scheme, to facilitate linear operations on ciphertexts. The HE scheme has 4 functions: `KeyGen`, `Enc`, `Dec`, and `Eval`. `KeyGen` generates a public key $pk$ and a secret key $sk$. `Enc` encrypts a message $m$ with the public key $pk$ to yield a ciphertext $c$. `Dec`, with the secret key $sk$ and ciphertext $c$ as inputs, decrypts the ciphertext to recover the message $m$. Finally, `Eval`, when given the public key $pk$, two ciphertexts $c_1$ and $c_2$ encrypting messages $m_1$ and $m_2$, along with a linear function $\mathcal{F}$, produces a new ciphertext $c'$ encrypting the result of $\mathcal{F}(m_1, m_2)$.
- **OT**. We use OT for non-linear operations (Rathee et al., 2020; 2021; Xue et al., 2023) in a network model. Specifically, we employ 1-out-of-2 correlated OT (Asharov et al., 2013) (2-COT$_\ell$) and 1-out-of-$k$ (Kolesnikov & Kumaresan, 2013) ($k$-OT$_\ell$). In 2-COT$_\ell$, the protocol takes as inputs the sender's correlation $x \in \mathbb{Z}_\ell$ and receiver's bit choice $i \in \{0, 1\}$. It then produces a random element $r \in \mathbb{Z}_\ell$ for the sender and $r + i \cdot x$ for the receiver. In $k$-OT$_\ell$, the sender possesses $k$ messages $m_0, ... m_{k-1}$ and the receiver holds an index $i \in [k]$. The protocol ensures the receiver learns $x_i$ as the output without learning any information about $x_j$, where $j \in [k]$ and $j \neq i$, while the sender learns nothing about the receiver's choice $i$.

**Prior Private Transformer Inference**. In response to the success of Transformers and the need to safeguard data privacy, various private Transformer Inferences (Chen et al., 2022; Zheng et al., 2023; Hao et al., 2022; Li et al., 2022; Lu et al., 2025; Hou et al., 2023; Luo et al., 2024; Pang et al., 2024) are proposed. To efficiently run private Transformer inferences, multiple cryptographic primitives are

used in a popular hybrid HE/MPC method IRON (Hao et al., 2022), i.e., in a Transformer, HE and SS are used for linear layers, and SS and OT are adopted for nonlinear layers. IRON and BumbleBee (Lu et al., 2025) focus on optimizing linear general matrix multiplications; SecFormer (Luo et al., 2024) improves non-linear operations, such as the exponential function, through polynomial approximation. BOLT (Pang et al., 2024) introduces the baby-step giant-step (BSGS) algorithm to reduce the number of HE rotations, proposes a word elimination (W.E.) technique, and uses polynomial approximation for non-linear operations, ultimately achieving state-of-the-art (SOTA) performance. It's worth noting that the W.E. technique in BOLT is not input-specific, as it uniformly prunes half the tokens regardless of the input or task. This approach may fail to remove all redundancy when it exceeds half of the tokens and can harm accuracy when redundancy is less than half. Additionally, since W.E. performs one-time pruning at the first layer rather than progressive, layer-by-layer pruning, it is less effective at reducing tokens for longer inputs. Moreover, BOLT's W.E. protocol is computationally expensive due to its reliance on sorting, whereas our method achieves lower asymptotic complexity and faster concrete runtime. Specifically, the state-of-the-art BOLT still faces efficiency and scalability challenges with long-token inputs. For example, one private inference with a GPT2-Base model can take $\sim 10$ minutes for 128-token inputs and $\sim 1$ hour for 512-token inputs, requiring data exchanges of more than 60GB and 200GB, respectively. Besides the hybrid HE/MPC methods, a line of works has explored performing private Transformer inference with only HE (Zimerman et al., 2023; Zhang et al., 2024). We leave a more detailed review of related works in Appendix G.

**Threat Model and Security Guarantee**. CipherPrune operates in a common private inference scenario where server $P_0$ owns a proprietary Transformer-based model $\mathcal{M}$ with private weights $w$, and client $P_1$ possesses private input data $x$. We assume the server and client are semi-honest, i.e., the server and client follow the designed protocols but are curious and attempt to learn extra information (e.g., $x$ or $w$). This setting is practical, as the server is incentivized to follow protocols and provide high-quality services for monetary gain, while the client is motivated to adhere to the protocol to receive those services. Consequently, this semi-honest setting is commonly adopted in existing works (Rathee et al., 2020; Huang et al., 2022; Hao et al., 2022; Lu et al., 2025; Pang et al., 2024). In this semi-honest setting, our protocols prevent the server from learning the client's data and the inference result; meanwhile, these protocols also block the client from accessing the model's parameters. In our protocols, we assume that both the server and client are aware of the number of pruned tokens. We argue that this information does not compromise the client's data or inference results, nor does it enable the client to access the model's weight parameters. Attacks that deviate from the semi-honest setting are beyond the scope of this work.

## 3 CipherPrune Framework

### 3.1 Motivation

Although plaintext pruning methods (Goyal et al., 2020; Kim & Cho, 2020; Wang et al., 2021) enable efficient and scalable inference for standard plaintext-domain Transformers, integrating these techniques into private Transformers remains challenging. Additionally, encrypted polynomial reduction and its joint optimization with token pruning remain largely unexplored.

**Challenge 1: Lacking protocols for input-specific, progressive encrypted token pruning and encrypted polynomial reduction.** The efficiency and scalability of plaintext pruning methods (Goyal et al., 2020; Kim & Cho, 2020; Wang et al., 2021) rely on two important features: (1) Input-specific pruning: This involves assigning an adaptive pruning ratio based on the dynamic importance of each input, as different inputs exhibit varying levels of redundancy. A fixed pruning ratio applied universally can result in suboptimal pruning or excessive pruning, leading to catastrophic accuracy loss. (2) Progressive pruning: This approach prunes tokens layer by layer, rather than performing a one-time early-layer pruning. Early pruning may fail to correctly identify redundant tokens, resulting in suboptimal or incorrect pruning. As mentioned earlier, prior W.E. in BOLT (Pang et al., 2024) is not input-specific or progressive. Furthermore, there is an absence of an encrypted protocol for polynomial production aimed at reducing the overhead of non-linear approximations.

**Challenge 2: Lacking network optimization to support joint token pruning and polynomial reduction.** In plaintext-domain token pruning (Goyal et al., 2020; Kim & Cho, 2020; Wang et al., 2021), there is no strong motivation to use polynomials to approximate non-linear activations, as these operations are straightforward and inexpensive to compute. However, in the ciphertext domain,

large-degree polynomials are employed to approximate non-linear functions for both efficiency and accuracy. We observed that polynomials of varying degrees can be assigned to different tokens in the encrypted domain based on their importance scores, which can also be optimized alongside token pruning. For instance, joint optimization of polynomial reduction and token pruning is essential, and network optimization involves searching for the optimal pruning and reduction thresholds.

**CipherPrune Overview.** In this paper, we introduce CipherPrune for an efficient and scalable private Transformer inference. Figure 3 shows the overview of Cipher-Prune. We first propose encrypted token pruning for both linear and non-linear activations in Section 3.2. Then, we develop an encrypted polynomial reduction for efficient non-linear operations in Section 3.3. We also introduce a network optimization for the joint optimization of token pruning and polynomial reduction in Section 3.4.

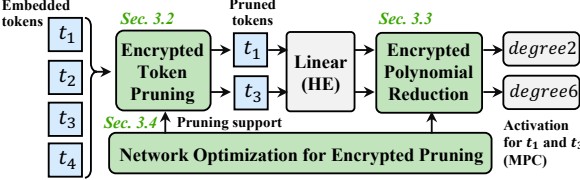

Figure 3: Overview of CipherPrune.

Figure 4 shows the workflow of a private Transformer inference implementation with our CipherPrune. During the private inference, ❶ the client's input is encrypted and multiplied with the embedding matrix in the server by the $\Pi_{MatMul}$ protocol. Then the result will be added to the positional encoding. ❷ The server performs the private attention computations, including linear projection via $\Pi_{MatMul}$ and non-linear attention map via $\Pi_{SoftMax}$, respectively. After obtaining the tokens and attention maps, ❸ our proposed encrypted token pruning method is performed to calculate token importance scores and compare these scores with the pruning threshold $\theta$ and reduction threshold $\beta$ to decide which tokens will be pruned or reduced in a privacy-preserving manner. The pruned tokens are discarded such that the following layers will have fewer operations. Low-degree polynomials are used to compute the GELU and SoftMax functions on reduced tokens. ❹ Layernorm and Feedforward operations will be executed via the prior protocols, $\Pi_{LayerNorm}$, $\Pi_{MatMul}$ and $\Pi_{GeLU}$. We detail our proposed encrypted token pruning method in the subsequent subsections.

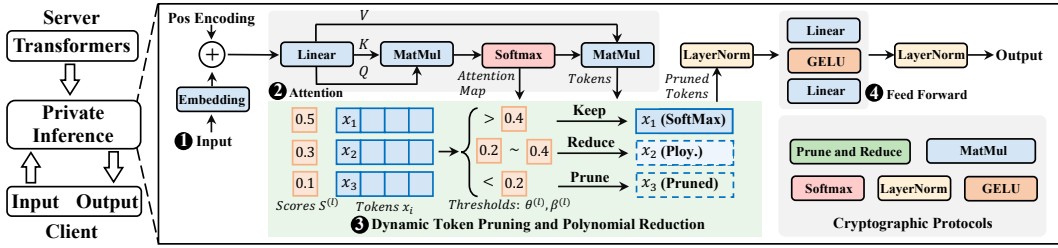

Figure 4: The workflow of a private Transformer inference with CipherPrune.

## 3.2 ENCRYPTED TOKEN PRUNING

**Pruning protocol $\Pi_{prune}$.** In private inference, confidentially pruning tokens presents a challenge, as the server and client must share attention maps and inputs without accessing their actual values. As depicted in Figure 5, during inference, attention maps are protected using ASS, with each party only able to view their respective shares.

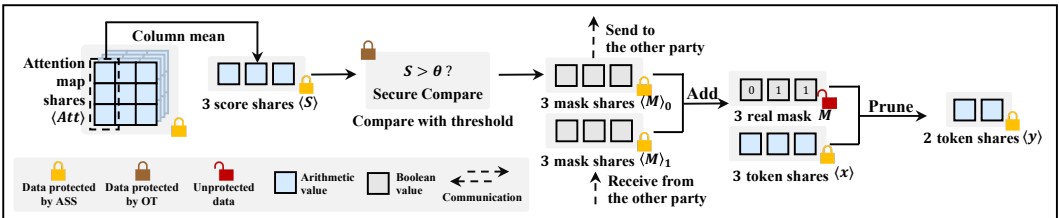

Figure 5: Illustration of mask generation and token pruning in $\Pi_{prune}$ with a non-sharing mask.

Specifically, the proposed secure token pruning protocol takes the secret-shared attention maps $\langle Att \rangle^h$ and tokens $\langle x \rangle$ as inputs, and outputs the pruned tokens $\langle y \rangle$ in a secret-sharing format. The secret shares are held by server $P_0$ and client $P_1$, respectively. First, $P_0$ and $P_1$ compute the importance score on their local secret shares, respectively. As depicted in Equation 1, the computation of

the importance score involves only addition and constant multiplication, which can be performed efficiently via ASS. After $P_0$ and $P_1$ acquire their respective shares of the importance score $\langle S \rangle$, they initiate a comparison protocol $\Pi_{CMP}$. This protocol contrasts the importance score against the threshold $\theta$ learned offline, enabling them to determine the shares of the resultant mask $\langle M \rangle$, where $M$ is 1 if $S > \theta$, otherwise 0.

Possessing the shares $\langle M \rangle$ without access to their real values prevents the direct pruning of tokens $\langle x \rangle$. A feasible solution involves reconstructing the non-shared mask, allowing both parties to independently prune their shares of the input sequence of $n$ tokens $\langle x \rangle$. This process then enables them to obtain the shares of the pruned output sequence of $m$ tokens $\langle y \rangle$. Appendix A, Figure 13 includes a more detailed and formal definition of $\Pi_{prune}$.

The overhead for our secure token pruning protocol is minimal. The importance score can be computed directly on shares, taking only 0.1 ms per attention module. This is efficient even for large models like BERT-Large, which has 24 heads per layer. Additionally, our protocol only requires $n$ invocations of the comparison protocol $\Pi_{CMP}$, each consistently completed within 5 ms, independent of the number of heads or embedding dimension. Thus, the total time complexity of our pruning protocol $\Pi_{prune}$ is linear, $O(n)$, based on the input token sequence length $n$.

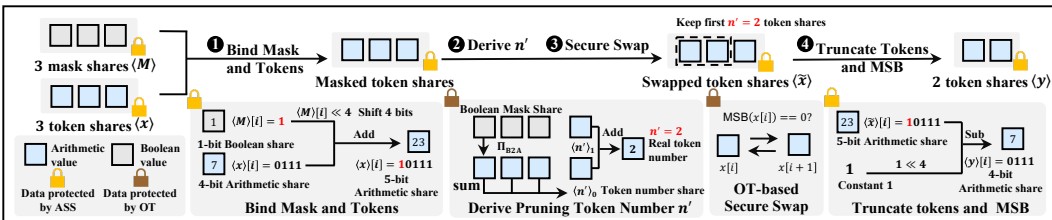

Figure 6: Example of token pruning with a protected mask.

**Pruning Mask Protocol $\Pi_{mask}$.** To further safeguard the privacy of the binary pruning mask $M$, specifically to protect the locations of pruned tokens, we design an additional Pruning Mask Protocol, $\Pi_{mask}$. The goal is to ensure that both parties, $P_0$ and $P_1$, can obtain the pruned token sequence without knowing which specific tokens were pruned. One key observation is that the number of tokens after pruning, $n'$, can be safely disclosed, as it is essential for subsequent processing and is not typically associated with significant security risks like adversarial attacks (Cui et al., 2021). Knowing $n'$ is crucial because token pruning involves relocating the less-important $m = n - n'$ tokens to the end of the token sequence while maintaining the order of the remaining $n'$ tokens. After this rearrangement, one can simply discard the $m$ ASS tokens at the end of the token list.

Figure 6 shows the secure mask protocol $\Pi_{mask}$ that is used to ensure the mask privacy in $\Pi_{prune}$. The protocol takes secret-shared token sequence $\langle x \rangle$ and mask $\langle M \rangle$ as inputs, and generates the pruned tokens $\langle y \rangle$. ❶ **Bind Mask and Tokens.** To swap tokens, their corresponding links with the mask will be disrupted. To preserve these links, there are two methods: one is to swap the masks and tokens respectively and simultaneously; the other is to bind the mask and tokens together so that they can be swapped as a unit. We adopt the second method, as binding the mask and tokens together proves more efficient than managing separate swaps for the mask and tokens. The bounded tokens $\langle \bar{x} \rangle$ can be obtained via left-shifting $\langle M \rangle$ by $f$ bits and adding to $\langle x \rangle$, where $f$ is the bit width of a token in $x$. Figure 6 illustrates using $f = 4$ as an example; however, in practice, $f$ can be flexibly adjusted. ❷ **Derive Pruning Token Number $n'$.** We found that $n'$ can be obtained by securely counting the number of 1s in $M$, which does not reveal the locations of 1s in $M$. Specifically, to determine $n'$, both $P_0$ and $P_1$ first convert their boolean mask shares $\langle M \rangle$ into a fixed-point format using $\Pi_{B2A}$. Each party then locally computes the sum of the arithmetic mask using ASS, yielding $\langle n' \rangle$. Finally, $P_0$ and $P_1$ obtain $n'$ by summing their respective shares, $\langle n' \rangle_0$ and $\langle n' \rangle_1$.

$$Swap(\bar{x}[i], \bar{x}[i+1]) = \begin{cases} b \cdot \bar{x}[i] + (1-b) \cdot \bar{x}[i+1], \\ b \cdot \bar{x}[i+1] + (1-b) \cdot \bar{x}[i]. \end{cases} \tag{2}$$

❸ **Secure Swap.** This step aims to enable $P_0$ and $P_1$ to iteratively move $m$ tokens to the end of the token sequence via OT-based oblivious swap defined in Equation 2. In each iteration, $P_0$ and $P_1$ perform an oblivious swap through the token sequence. To privately swap two tokens $\bar{x}[i]$ and $\bar{x}[i+1]$, they first extract the MSB $b$ from the bounded token $\bar{x}[i]$ and perform four OT-based multiplications.

❹ **Truncate Tokens and MSB.** $P_0$ and $P_1$ can truncate the swapped token sequence $\langle \tilde{x} \rangle$ and remove the MSB respectively to obtain the pruned token sequence. Figure 14 in the Appendix A details more about the mask protection protocol.

**Analysis.** The complexity of the proposed $\Pi_{mask}$ mainly depends on the number of oblivious swaps. To prune $m$ tokens out of $n$ input tokens, $O(mn)$ swaps are needed. Since token pruning is performed progressively, only a small number of tokens are pruned at each layer, which makes $\Pi_{mask}$ efficient during runtime. Specifically, for a BERT-Base model with 128 input tokens, the pruning protocol only takes $\sim 0.9$s on average in each layer.

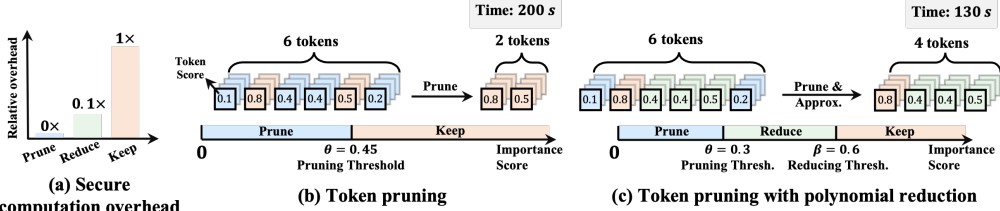

Figure 7: Comparison of token pruning-only method and pruning with polynomial reduction.

## 3.3 ENCRYPTED POLYNOMIAL REDUCTION

After pruning, retained tokens still require expensive operations, particularly for costly non-linear functions. These non-linear functions are computed via expensive high-degree polynomials (Lu et al., 2025; Pang et al., 2024). We notice that we can reduce the high-degree polynomials to their low-degree counterparts for the less important tokens. As demonstrated in Figure 7 (a), the cost of reduced polynomial can be $0.1\times$ that of the high-degree polynomial. This motivates us to accelerate the non-linear operations with low-degree polynomials while maintaining the desired accuracy. Similar to employing a threshold $\theta$ to prune tokens with importance scores below $\theta$, we use another reduction threshold $\beta$ ($\beta > \theta$) to identify tokens for reduction. As shown in Figure 7 (b)(c), combining token pruning with polynomial reduction further reduces execution time compared to the pruning-only method. Importantly, we can optimize both $\theta$ and $\beta$ together during offline fine-tuning to enhance efficiency. During the online inference phase, polynomial reduction occurs after token pruning (the pruning ratio greater than zero). This simplifies the reduction process: the tokens have already been pruned, and the locations of these tokens are rotated and concealed. Consequently, there's no need to safeguard the token mask for reduction. Instead, we can simply modify the pruning protocol $\Pi_{Prune}$ to establish the reduction protocol. As illustrated in Figure 8, the pruned tokens are determined by executing protocols $\Pi_{prune}$ and $\Pi_{mask}$ in tandem. A secure com-

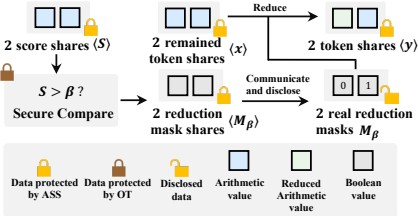

Figure 8: Secure polynomial reduction.

parison with the reduction threshold $\beta$ then produces the reduction mask $\langle M_\beta \rangle$. The location of this mask corresponds to pruned tokens, not the original tokens, so revealing it does not compromise the location privacy of reduced tokens, provided that the privacy of pruned locations is maintained. Once the reduction mask $M_\beta$ is known to each party, it can be used to guide the decision on whether to apply the high-degree polynomials or low-degree ones for the non-linear functions, where 1 indicates using the high-degree ones and 0 signifies low-degree ones. The choice of these approximation polynomials can be flexible. We utilize prior non-linear function approximation methods as detailed in references such as (Kim et al., 2021; Lu et al., 2025; Pang et al., 2024). The specific configurations used are outlined in the Appendix C.

## 3.4 NETWORK OPTIMIZATION FOR ENCRYPTED PRUNING AND POLYNOMIAL REDUCTION

To support online token pruning and non-linear approximation, an offline fine-tuning method is needed to optimize pruning and approximation thresholds, $\theta$ and $\beta$, to minimize inference overhead and achieve user-defined accuracy above $a$. This process is challenging because (1) previous studies have not incorporated the efficiency and accuracy of encrypted token pruning and non-linear approximation into the fine-tuning phase, and (2) the thresholds for different Transformer layers vary and are

---

**Algorithm 1** Crypto-aware Thresholds Learning

---

Input: pre-trained Transformer $\mathcal{M}$, training data $\mathcal{D}$, initial thresholds $\theta, \beta$

1. Set model $\mathcal{M}$ with $L$ layers, weights $w$, input tokens $x$, accuracy requirement $a$, and hyperparameters $T, \lambda, \alpha$.
2. Search for optimal thresholds $\theta$, $\beta$ and weights $w$ on data $\mathcal{D}$.

    (a) For $l \in [L]$, set soft masks for token $x_i$, $M_\theta^{(l)}(x_i) = \sigma(\frac{S^{(l)}(x_i) - \theta^{(l)}}{T})$ and $M_\beta^{(l)}(x_i) = \sigma(\frac{S^{(l)}(x_i) - \beta^{(l)}}{T})$.

    (b) For $l \in [L]$, integrate the crypto-friendly polynomial activation functions. In the $l$-th layer, compute GELU function as: $\mathsf{GELU}(x_i) = M_\beta^{(l)}(x_i) \cdot \mathsf{GELU}(x_i) + (1 - M_\beta^{(l)}(x_i)) \cdot \mathsf{ApproxGELU}(x_i)$. Except for the first layer, compute SoftMax function as: $\mathsf{SoftMax}(x_i) = M_\beta^{(l-1)}(x_i) \cdot \mathsf{SoftMax}(x_i) + (1 - M_\beta^{(l-1)}(x_i)) \cdot \mathsf{ApproxSoftMax}(x_i)$. For the input feature $x_{in}$, compute the output feature $x_{out}$ in the $l$-th layer as: $x_{out} = M_\theta^{(l)}(x_{in}) \cdot x_{out}$

    (c) Update $w$, $\theta$ and $\beta$ jointly to minimize the loss $\mathcal{L}$, where $\mathcal{L} = \mathcal{L}_{task} + \lambda(\mathcal{L}_{prune} + \alpha\mathcal{L}_{approx.})$.

3. Finetune $w$ on data $\mathcal{D}$ with learned thresholds $\theta$ and $\beta$.

    (a) Fix the threshold $\theta$, $\beta$. Binarize the mask $M_\theta^{(l)}$ in every layer as:
$$M_\theta^{(l)}(x_i) = \begin{cases} 1 & \text{if } S^{(l)}(x_i) > \theta^{(l)}, \\ 0 & \text{otherwise.} \end{cases}, \text{ and } M_\beta^{(l)} \text{ is binarized in the same way.}$$

    (b) Update $w$ to minimize the loss $\mathcal{L}_{task}$. Derive the optimal fine-tuned transformer $\mathcal{M}^*$.

4. Output $\theta$, $\beta$ and $w$ if accuracy $\geq a$; otherwise back to step 2.

---

challenging to pinpoint. To address these challenges, we introduce a crypto-aware fine-tuning method, outlined in Algorithm 1. This method uses a gradient search approach and proposes to incorporate crypto-aware pruning and approximation into the training phase. Additionally, the loss functions are designed to optimize both efficiency and accuracy.

After initialization, we make the masks differentiable during fine-tuning to allow for trainable thresholds, as shown in step 2.(a) of Algorithm 1. Here, $T$ represents the temperature, and $\sigma$ is the Sigmoid function. This soft mask, a differentiable approximation of the binary mask, enables gradient-based updates to $\theta$ and $\beta$. In step 2.(b) of the same algorithm, we introduce polynomial activation functions during the fine-tuning phase. ApproxSoftMax and ApproxGELU are low-degree polynomial approximations of the SoftMax and GELU functions. The ApproxSoftMax replaces the exponential function $e^x$ in original SoftMax with a Taylor series $(1 + \frac{x}{2^n})^{2^n}$. And the ApproxGELU leverages simple polynomials such as $p^3(x) = -0.51 - 0.42x^2 + -0.12x^2 - 0.01x^3$ to approximate GELU. We defer the detailed polynomial in Equations 5 and 8 in the Appendix C. If a token $x_i$'s importance score exceeds the threshold $\beta$, it activates mainly through the original SoftMax or GELU functions; otherwise, through their polynomial approximations.

$$\mathcal{L}_{prune} = \frac{1}{L}\sum_{l=0}^{L-1} \left\| M_\theta^{(l)}(x) \right\|_1, \mathcal{L}_{approx.} = \frac{1}{L}\sum_{l=0}^{L-1} \left\| M_\beta^{(l)}(x) \right\|_1 \tag{3}$$

The overall objective function is designed to minimize the loss function $\mathcal{L} = \mathcal{L}_{task} + \lambda(\mathcal{L}_{prune} + \alpha\mathcal{L}_{approx.})$ where $L$ denotes the number of layers. $\mathcal{L}_{task}$ optimizes accuracy for downstream tasks, while $\mathcal{L}_{prune}$ and $\mathcal{L}_{approx.}$, defined by $M_\theta$'s and $M_\beta$'s $l_1$-norms respectively, target efficiency as detailed in Equation 3. The hyperparameters $\lambda$ and $\alpha$ dictate the extent of pruning and approximation, with higher values leading to increased pruning or approximation. This structure introduces additional gradients, pushing $\theta$ and $\beta$ towards minimizing $\mathcal{L}_{prune}$ and $\mathcal{L}_{approx.}$. Once the optimized $\theta$ and $\beta$ are determined, we fix them and proceed to fine-tune the model weights to meet the accuracy requirement $a$ as specified in Step 3. For each Transformer layer, we binarize the masks $M_\theta$ and $M_\beta$ to select tokens for pruning or approximation. Subsequently, we update the model weights to minimize the downstream task loss $\mathcal{L}_{task}$.

# 4 EXPERIMENTS

## 4.1 EXPERIMENTAL SETUP

**Models and Datasets**. We evaluated CipherPrune on the GPT2-Base and three BERT variants (Devlin et al., 2018): BERT-Medium, BERT-Base, and BERT-Large. These models are commonly used in private Transformer frameworks. Similar to prior work (Pang et al., 2024), we fine-tune the BERT models on four downstream NLP tasks in GLUE benchmarks (Wang et al., 2018): the Multi-Genre Natural Language Inference Corpus (MNLI), the Stanford Question Answering Dataset (QNLI), the Stanford Sentiment Treebank (SST-2), and the Microsoft Research Paraphrase Corpus (MRPC).

**System Setup and Implementation**. We encode floating-point parameters in Transformers into fixed-point numbers and set the scale according to prior works(Hao et al., 2022; Lu et al., 2025; Pang et al., 2024). CipherPrune uses the EzPC (EzP, 2023) framework and the SEAL (SEA, 2023) library. EzPC compiles TensorFlow-based deep neural networks into secure computation protocols running on cryptographic backends. We simulate LAN with 3Gbps bandwidth and 0.8ms ping, and WAN with 200Mbps bandwidth and 40ms ping, following (Pang et al., 2024). All experiments are conducted on an AMD Ryzen Threadripper PRO 3955WX (2.2GHz, 125GB RAM) and fine-tuning of the BERT model with threshold learning is done on NVIDIA GeForce RTX 3090 GPUs with CUDA 11.0.3.

## 4.2 RESULTS

Table 1: End-to-end comparison of CipherPrune with prior works on BERT models. Time is in seconds. Comm. stands for communication in GB and Acc. for accuracy in percentage.

| Method | BERT Medium | | | BERT Base | | | BERT Large | | |
|---|---|---|---|---|---|---|---|---|---|
| | Time | Comm. | Acc. | Time | Comm. | Acc. | Time | Comm. | Acc. |
| IRON (Hao et al., 2022) | 442.4 | 124.5 | $87.7_{\pm 0.2}$ | 1087.8 | 281.0 | $90.4_{\pm 0.1}$ | 2873.5 | 744.8 | $92.7_{\pm 0.1}$ |
| BOLT w/o W.E. (Pang et al., 2024) | 197.1 | 27.9 | $87.4_{\pm 0.3}$ | 484.5 | 59.6 | $90.3_{\pm 0.1}$ | 1279.8 | 142.6 | $92.6_{\pm 0.2}$ |
| BOLT (Pang et al., 2024) | 99.5 | 14.3 | $87.2_{\pm 0.3}$ | 245.4 | 25.7 | $89.9_{\pm 0.3}$ | 624.3 | 67.9 | $92.4_{\pm 0.2}$ |
| CipherPrune | 43.6 | 6.7 | $87.4_{\pm 0.2}$ | 79.1 | 9.7 | $90.1_{\pm 0.2}$ | 157.6 | 18.4 | $92.5_{\pm 0.1}$ |

**End-to-end performance.** In Table 1, we evaluate CipherPrune on three BERT models, comparing it with previous private Trasnformer frameworks: IRON (Hao et al., 2022) and BOLT (Pang et al., 2024). CipherPrune achieves up to $\sim 18.2\times$ speedup over IRON on the BERT-Large model and $\sim 8.1\times$ speedup over vanilla BOLT without W.E.. When compared with BOLT with the word elimination technique, CipherPrune is still $\sim 3.9\times$ faster without compromising accuracy. Communication costs are also reduced by $2.3 \sim 40.4\times$ compared to prior works. Compared with BOLT, CipherPrune can remove more redundant tokens during inference thorough the adaptive and progressive pruning strategy. Moreover, CipherPrune also leverages low-degree polynomials to further reduce the computation and communication overhead. CipherPrune can easily extend to other frameworks. Comparison with more related works like BumbleBee (Lu et al., 2025), MPCFormer (Li et al., 2022) and PUMA (Dong et al., 2023) can be found in Appendix D.

Table 2: Accuracy and time comparisons of different methods. CipherPrune[†] stands for CipherPrune with token pruning only.

| Method | Accuracy Metric on Tasks (%) | | | | Time(Sec) |
|---|---|---|---|---|---|
| | MNLI | QNLI | SST2 | MPRC | |
| BOLT w/o W.E. | 84.75 | 90.32 | 91.74 | 90.53 | 484.5 |
| BOLT | 84.71 | 89.94 | 92.74 | 89.95 | 245.4 |
| CipherPrune[†] | 84.74 | 90.17 | 92.75 | 90.45 | 115.3 |
| CipherPrune | 84.68 | 90.11 | 92.66 | 90.18 | 79.1 |

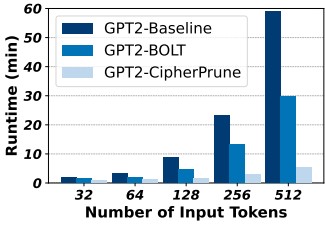

Figure 9: Runtime on GPT2.

**Token pruning and polynomial reduction.** Table 2 demonstrates the the effects of the main design blocks in CipherPrune: adaptive token pruning and polynomial reduction. Our baseline is the vanilla BOLT framework without W.E.. BOLT's W.E. removes $50\%$ of the input tokens and effectively cuts the overhead of cryptographic protocols by half. With fine-tuning, the W.E. incurs only marginal accuracy loss. Yet, the adaptive and progressive token pruning in CipherPrune[†] can further improve the utility-accuracy trade-off. Instead of setting the pruning ratio as $50\%$ manually, CipherPrune[†] adaptively decides the pruning ratio based on both the input length and content. This contributes to up to $0.5\%$ better accuracy. On the other hand, the progressive pruning in CipherPrune[†] allows to remove more redundant information, contributing to $2.1\times$ runtime speed up over BOLT with W.E.. By incorporating polynomial reduction, CipherPrune can achieve up to $6.1\times$ speed up over BOLT. While the accuracy drops slightly from CipherPrune[†], it is still comparable or even higher than BOLT.

**Scalability with the input length.** In Figure 9, we compare the runtime of CipherPrune and BOLT with varying input token numbers on GPT2. The baseline is BOLT without W.E.. The quadratic complexity of Transformer inference makes it challenging for BOLT to scale to long inputs. Although W.E. can reduce the overhead of private inference by half, BOLT with W.E. still scales quadratically

with the number of input tokens. In contrast, CipherPrune demonstrates increasingly significant runtime savings as the input length grows. With 32 input tokens, CipherPrune achieves a $\sim 1.9\times$ speedup. When the input length reaches 512 tokens, CipherPrune is $\sim 10.6\times$ faster than the baseline. The details about private inference on GPT2 models can be found in Appendix D.

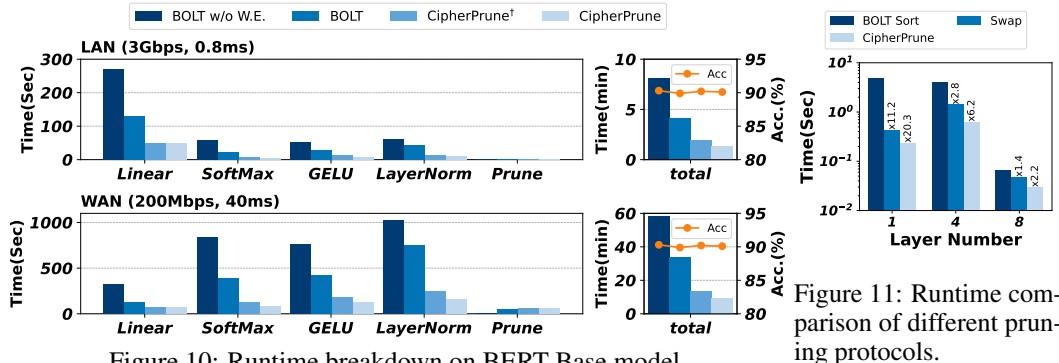

Figure 10: Runtime breakdown on BERT-Base model.

Figure 11: Runtime comparison of different pruning protocols.

**Runtime breakdown.** In Figure 10, we break down the runtime for each protocol in the BERT-Base model with 128 input tokens. In the LAN setting, the communication is efficient and the main bottleneck is the HE-based linear operation. In contrast, the massive communication of the non-linear operations becomes the bottleneck in the WAN setting. Since pruned tokens are excluded from the computation in all subsequent layers, CipherPrune can effectively reduce the overhead of both linear and non-linear operations. This contributes to CipherPrune's efficiency in both LAN setting and WAN setting. As shown in Figure 10, the proposed pruning protocols in CipherPrune are lightweight, accounting for only $1.6\%$ of the total runtime. This is because $\Pi_{prune}$ leverages ASS to offload substantial computation to the local side, such as accumulating the importance score. Additionally, $\Pi_{mask}$ utilizes the number of tokens in each layer to avoid sorting the whole token sequence.

**Analysis on different pruning protocols.** As shown in Figure 11, we compare the efficiency of different pruning protocols. BOLT's W.E. uses Bitonic sort to sort the whole token sequence, which needs $O(n \log^2 n)$ oblivious swaps. In CipherPrune, the client and server only need $O(mn)$ oblivious swaps to relocate and prune the less important tokens. Since only a small number of tokens are removed in each layer, CipherPrune has a linear complexity to $n$ in general. By binding the mask with tokens on the MSB, CipherPrune can handle the token sequence and pruning mask in one go and achieves $2.2 \sim 20.3\times$ speed up.

**Study on the pruning parameters.** In Figure 12, we show the accuracy-latency trade-off for the BERT-Base model under different parameter settings. Larger $\lambda$ and $\alpha$ result in more tokens being pruned or reduced. With $\lambda$ less than 0.05, an appropriate ratio of tokens is pruned, maintaining a stable accuracy of around 90%. Smaller $\alpha$ leads to more tokens being computed with high-degree polynomials, which increases accuracy but also latency. Notably, accuracy with a large $\alpha$ is higher than with a large $\lambda$. This is because many tokens are reduced but not discarded, preserving necessary information for accurate inference.

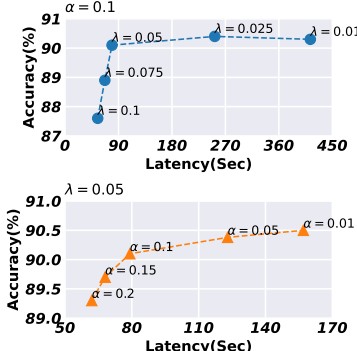

Figure 12: Ablation study on hyperparameters $\lambda$ and $\alpha$.

## 5 CONCLUSION

The proposed CipherPrune addresses the critical efficiency and scalability challenges of private Transformer inference by introducing a novel approach that combines encrypted token pruning and polynomial reduction protocols. By progressively pruning redundant tokens and reducing the polynomial degree for less important tokens, CipherPrune significantly reduces runtime overhead while maintaining accuracy. Our experiments confirm its effectiveness, achieving a substantial reduction in execution time compared to previous methods.

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

APPENDIX

# A  SECURE TOKEN PRUNING PROTOCOLS

We detail the encrypted token pruning protocols $\Pi_{prune}$ in Figure 13 and $\Pi_{mask}$ in Figure 14 in this section.

---

**Parties:** Server $P_0$, Client $P_1$.
**Input:** $P_0$ and $P_1$ holds $\{\langle Att\rangle_0^h, \langle Att\rangle_1^h\}_{h=0}^{H-1} \in \mathbb{Z}_{2^\ell}^{n\times n}$ and $\langle x\rangle_0, \langle x\rangle_1 \in \mathbb{Z}_{2^\ell}^{n\times D}$ respectively, where H is the number of heads, n is the number of input tokens and D is the embedding dimension of tokens. Additionally, $P_1$ holds a threshold $\theta \in \mathbb{Z}_{2^\ell}$.
**Output:** $P_0$ and $P_1$ get $\langle y\rangle_0, \langle y\rangle_1 \in \mathbb{Z}_{2^\ell}^{n'\times D}$, respectively, where $y = \mathsf{Prune}(x)$ and $n'$ is the number of remaining tokens.

---

**Protocol:**

1: For $h \in [H]$, $P_0$ and $P_1$ compute locally with input $\langle Att\rangle^h$, and learn the importance score in each head $\langle s\rangle^h \in \mathbb{Z}_{2^\ell}^n$, where $\langle s\rangle^h[j] = \frac{1}{n}\sum_{i=0}^{n-1}\langle Att\rangle^h[i,j]$.

2: $P_0$ and $P_1$ compute locally with input $\{\langle s\rangle^i \in \mathbb{Z}_{2^\ell}^n\}_{i=0}^{H-1}$, and learn the final importance score $\langle S\rangle \in \mathbb{Z}_{2^\ell}^n$ for each token, where $\langle S\rangle[i] = \frac{1}{H}\sum_{h=0}^{H-1}\langle s\rangle^h[i]$.

3: For $i \in [n]$, $P_0$ and $P_1$ invoke $\Pi_{CMP}$ with inputs $\langle S\rangle$ and $\theta$, and learn $\langle M\rangle \in \mathbb{Z}_{2^\ell}^n$, such that$\langle M\rangle[i] = \Pi_{CMP}(\langle S\rangle[i] - \theta)$, where:
$M[i] = \begin{cases} 1 & \text{if } S[i] > \theta, \\ 0 & \text{otherwise.} \end{cases}$

4: $P_0$ and $P_1$ invoke $\Pi_{mask}$ with inputs $\langle x\rangle$ and pruning mask $\langle M\rangle$, and set outputs as $\langle y\rangle$.

---

Figure 13: Secure Token Pruning Protocol $\Pi_{prune}$.

---

**Parties:** Server $P_0$, Client $P_1$.
**Input:** $P_0$ and $P_1$ hold $\langle x\rangle_0, \langle x\rangle_1 \in \mathbb{Z}_{2^\ell}^{n\times D}$ and $\langle M\rangle_0, \langle M\rangle_1 \in \mathbb{Z}_{2^\ell}^n$, respectively, where n is the number of input tokens and D is the embedding dimension of tokens.
**Output:** $P_0$ and $P_1$ get $\langle y\rangle_0, \langle y\rangle_1 \in \mathbb{Z}_{2^\ell}^{n'\times D}$, respectively, where $y = \mathsf{Prune}(x)$ and $n'$ is the number of remaining tokens.

---

**Protocol:**

1: For $i \in [n]$, $P_0$ and $P_1$ set $\langle M\rangle$ to the MSB of $\langle x\rangle$ and learn the masked tokens $\langle \bar{x}\rangle \in Z_{2^\ell}^{n\times D}$, where $\langle \bar{x}[i]\rangle = \langle x[i]\rangle + (\langle M[i]\rangle << f)$ and $f$ is the fixed-point precision.

2: $P_0$ and $P_1$ compute the sum of $\{\Pi_{B2A}(\langle M\rangle[i])\}_{i=0}^{n-1}$, and learn the number of remaining tokens $n'$ and the number of tokens to be pruned $m$, where $m = n - n'$.

3: For $k \in [m]$, for $i \in [n-k-1]$, $P_0$ and $P_1$ invoke $\Pi_{msb}$ to learn the highest bit of $\langle \bar{x}[i]\rangle$, where $b = \mathsf{MSB}(\bar{x}[i])$. With the highest bit of $\bar{x}[i]$, $P_0$ and $P_1$ perform a oblivious swap between $\bar{x}[i]$ and $\bar{x}[i+1]$: $\begin{cases} \tilde{x}[i] = b\cdot\bar{x}[i] + (1-b)\cdot\bar{x}[i+1] \\ \tilde{x}[i+1] = b\cdot\bar{x}[i+1] + (1-b)\cdot\bar{x}[i] \end{cases}$
$P_0$ and $P_1$ learn the swapped token sequence $\langle \tilde{x}\rangle$.

4: $P_0$ and $P_1$ truncate $\langle \tilde{x}\rangle$ locally by keeping the first $n'$ tokens, clear current MSB (all remaining token has 1 on the MSB), and set outputs as $\langle y\rangle$.

---

Figure 14: Secure Mask Protocol $\Pi_{mask}$.

**Complexity of $\Pi_{mask}$.** The complexity of the proposed $\Pi_{mask}$ mainly depends on the number of oblivious swaps. To prune $m$ tokens out of $n$ input tokens, $O(mn)$ swaps are needed. Since token pruning is performed progressively, only a small number of tokens are pruned at each layer, which

makes $\Pi_{mask}$ efficient during runtime. Specifically, for a BERT base model with 128 input tokens, the pruning protocol only takes $\sim 0.9$s on average in each layer. An alternative approach is to invoke an oblivious sort algorithm (Bogdanov et al., 2014; Pang et al., 2024) on $\langle \bar{x} \rangle$. However, this approach is less efficient because it blindly sort the whole token sequence without considering $m$. That is, even if only 1 token needs to be pruned, $O(nlog^2 n) \sim O(n^2)$ oblivious swaps are needed, where as the proposed $\Pi_{mask}$ only need $O(n)$ swaps. More generally, for an $\ell$-layer Transformer with a total of $m$ tokens pruned, the overall time complexity using the sort strategy would be $O(\ell n^2)$ while using the swap strategy remains an overall complexity of $O(mn)$. Specifically, using the sort strategy to prune tokens in one BERT Base model layer can take up to $3.8 \sim 4.5$ s depending on the sorting algorithm used. In contrast, using the swap strategy only needs $0.5$ s. Moreover, alternative to our MSB strategy, one can also swap the encrypted mask along with the encrypted token sequence. However, we find that this doubles the number of swaps needed, and thus is less efficient the our MSB strategy, as is shown in Figure 11.

## B    EXISTING PROTOCOLS

**Existing Protocols Used in Our Private Inference.** In our private inference framework, we reuse several existing cryptographic protocols for basic computations. $\Pi_{MatMul}$ (Pang et al., 2024) processes two ASS matrices and outputs their product in SS form. For non-linear computations, protocols $\Pi_{SoftMax}, \Pi_{GELU}$, and $\Pi_{LayerNorm}$(Lu et al., 2025; Pang et al., 2024) take a secret shared tensor and return the result of non-linear functions in ASS. Basic protocols from (Rathee et al., 2020; 2021) are also utilized. $\Pi_{CMP}$(EzP, 2023), for example, inputs ASS values and outputs a secret shared comparison result, while $\Pi_{B2A}$(EzP, 2023) converts secret shared Boolean values into their corresponding arithmetic values.

## C    POLYNOMIAL REDUCTION FOR NON-LINEAR FUNCTIONS

The SoftMax and GELU functions can be approximated with polynomials. High-degree polynomials (Lu et al., 2025; Pang et al., 2024) can achieve the same accuracy as the LUT-based methods Hao et al. (2022). While these polynomial approximations are more efficient than look-up tables, they can still incur considerable overheads. Reducing the high-degree polynomals to the low-degree ones for the less important tokens can imporve efficiency without compromising accuracy. The SoftMax function is applied to each row of an attention map. If a token is to be reduced, the corresponding row will be computed using the low-degree polynomial approximations. Otherwise, the corresponding row will be computed accurately via a high-degree one. That is if $M'_\beta[i] = 1$, $P_0$ and $P_1$ uses high-degree polynomials to compute the SoftMax function on token $x[i]$:

$$\mathsf{SoftMax}_i(x) = \frac{e^{x_i}}{\sum_{j \in [d]} e^{x_j}} \tag{4}$$

where $x$ is a input vector of length $d$ and the exponential function is computed via a polynomial approximation. For the SoftMax protocol, we adopt a similar strategy as (Kim et al., 2021; Hao et al., 2022), where we evaluate on the normalized inputs $\mathsf{SoftMax}(x - max_{i \in [d]} x_i)$. Different from (Hao et al., 2022), we did not used the binary tree to find max value in the given vector. Instead, we traverse through the vector to find the max value. This is because each attention map is computed independently and the binary tree cannot be re-used. If $M_\beta[i] = 0$, $P_0$ and $P_1$ will approximate the SoftMax function with low-degree polynomial approximations. We detail how SoftMax can be approximated as follows:

$$\mathsf{ApproxSoftMax}_i(x) = \frac{\mathsf{ApproxExp}(x_i)}{\sum_{j \in [d]} \mathsf{ApproxExp}(x_j)} \tag{5}$$

$$\mathsf{ApproxExp}(x) = \begin{cases} 0 & \text{if } x \leq T \\ (1 + \frac{x}{2^n})^{2^n} & \text{if } x \in [T, 0] \end{cases} \tag{6}$$

where the $2^n$-degree Taylor series is used to approximate the exponential function and $T$ is the clipping boundary. The value $n$ and $T$ determines the accuracy of above approximation. With $n = 6$ and $T = -13$, the approximation can achieve an average error within $2^{-10}$ (Lu et al., 2025). For low-degree polynomial approximation, $n = 3$ is used in the Taylor series.

Similarly, $P_0$ or $P_1$ can decide whether or not to approximate the GELU function for each token. If $M_\beta[i] = 1$, $P_0$ and $P_1$ use high-degree polynomials (Lu et al., 2025) to compute the GELU function on token $x[i]$ with high-degree polynomial:

$$\text{ApproxGELU}(x) = \begin{cases} 0 & \text{if } x \leq -5 \\ P^3(x), & \text{if } -5 < x \leq -1.97 \\ P^6(x), & \text{if } -1.97 < x \leq 3 \\ x, & \text{if } x > 3 \end{cases} \tag{7}$$

where $P^3(x)$ and $P^6(x)$ are degree-3 and degree-6 polynomials respectively. The detailed coefficient for the polynomial is:

$$P^3(x) = -0.50540312 - 0.42226581x - 0.11807613x^2 - 0.01103413x^3$$

, and

$$P^6(x) = 0.00852632 + 0.5x + 0.36032927x^2 - 0.03768820x^4 + 0.00180675x^6$$

For BOLT baseline, we use another high-degree polynomial to compute the GELU function.

$$\text{ApproxGELU}(x) = \begin{cases} 0 & \text{if } x < -2.7 \\ P^4(x), & \text{if } |x| \leq 2.7 \\ x, & \text{if } x > 2.7 \end{cases} \tag{8}$$

We use the same coefficients for $P^4(x)$ as BOLT (Pang et al., 2024).

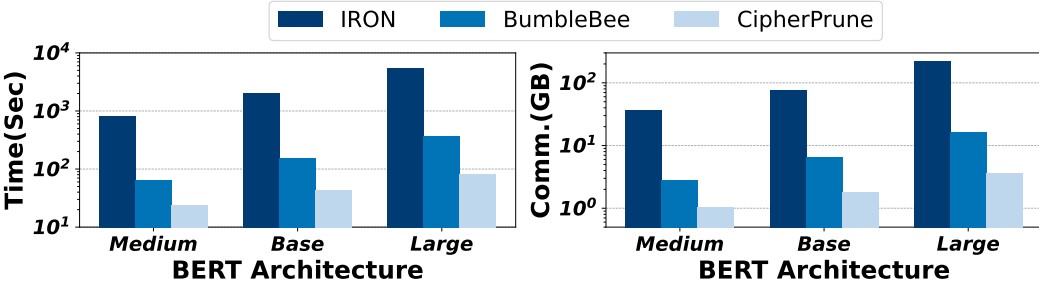

Figure 15: Comparison with prior works on the BERT model. The input has 128 tokens.

If $M'_\beta[i] = 0$, $P_0$ and $P_1$ will use low-degree polynomial approximation to compute the GELU function instead. Encrypted polynomial reduction leverages low-degree polynomials to compute non-linear functions for less important tokens. For the GELU function, the following degree-2 polynomial Kim et al. (2021) is used:

$$\text{ApproxGELU}(x) = \begin{cases} 0 & \text{if } x < -1.7626 \\ 0.5x + 0.28367x^2, & \text{if } x \leq |1.7626| \\ x, & \text{if } x > 1.7626 \end{cases}$$

## D  COMPARISON WITH MORE RELATED WORKS.

**Other 2PC frameworks.** The primary focus of CipherPrune is to accelerate the private Transformer inference in the 2PC setting. As shown in Figure 15, CipherPrune can be easily extended to other 2PC private inference frameworks like BumbleBee (Lu et al., 2025). We compare CipherPrune with BumbleBee and IRON on BERT models. We test the performance in the same LAN setting as BumbleBee with 1 Gbps bandwidth and 0.5 ms of ping time. CipherPrune achieves more than $\sim 60\times$ speed up over BOLT and $4.3\times$ speed up over BumbleBee.

**Extension to 3PC frameworks.** Additionally, we highlight that CipherPrune can be also extended to the 3PC frameworks like MPCFormer (Li et al., 2022) and PUMA (Dong et al.,

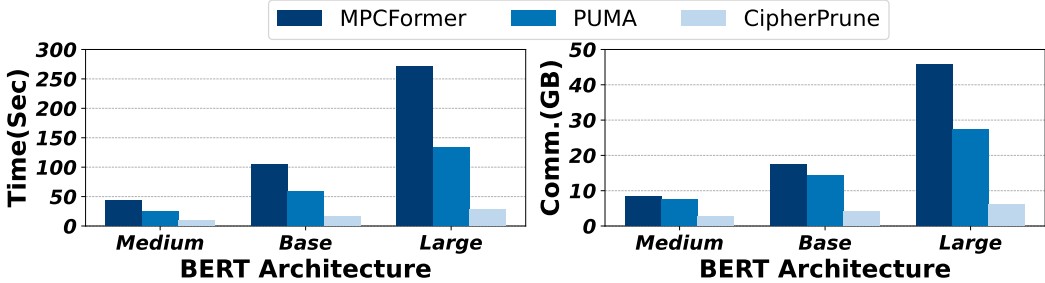

Figure 16: Comparison with MPCFormer and PUMA on the BERT models. The input has 128 tokens.

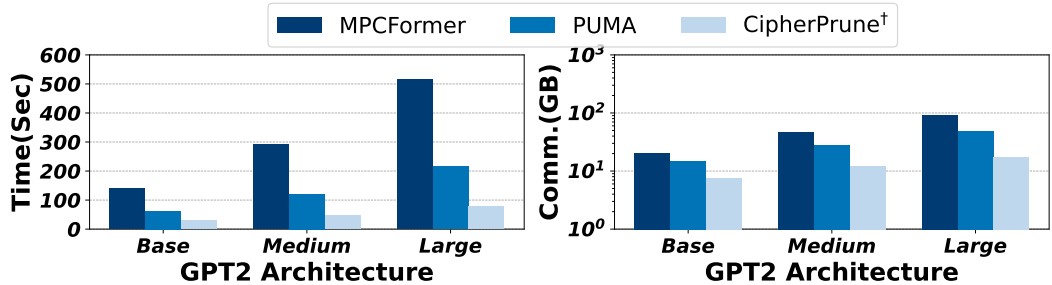

Figure 17: Comparison with MPCFormer and PUMA on the GPT2 models. The input has 128 tokens. The polynomial reduction is not used.

2023). This is because CipherPrune is built upon basic primitives like comparison and Boolean-to-Arithmetic conversion. We compare CipherPrune with MPCFormer and PUMA on both the BERT and GPT2 models. CipherPrune has a $6.6 \sim 9.4\times$ speed up over MPCFormer and $2.8 \sim 4.6\times$ speed up over PUMA on the BERT-Large and GPT2-Large models. For a fair comparison with prior works (Li et al., 2022; Dong et al., 2023), we report the runtime and communication cost measured from the prefilling phase. As shown in Figure 18, we perform private inference on GPT2 models to generate one token. Similar to PUMA, we do not consider the whole autoregressive decoding process. We note that the Key-Value (KV) cache and its

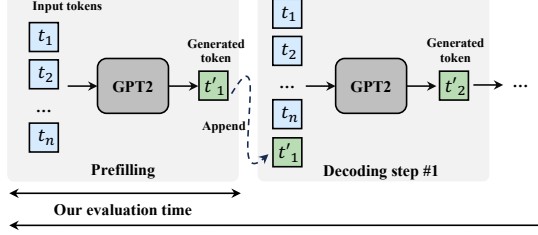

Figure 18: Illustration of evaluation on GPT2 models.

compression (Li et al., 2024) can further improve the efficiency of the decoding phase for GPT2 models. We leave private KV cache compression for future work.

## E COMMUNICATION REDUCTION IN SOFTMAX AND GELU.

Table 3: Communication cost (in MB) of the SoftMax and GELU protocol in each Transformer layer.

| Layer Index | 0 | 1 | 2 | 3 | 4 | 5 | 6 | 7 | 8 | 9 | 10 | 11 |
|---|---|---|---|---|---|---|---|---|---|---|---|---|
| Softmax | 642.19 | 642.19 | 642.19 | 642.19 | 642.19 | 642.19 | 642.19 | 642.19 | 642.19 | 642.19 | 642.19 | 642.19 |
| Pruned Softmax | 642.19 | 129.58 | 127.89 | 119.73 | 97.04 | 71.52 | 43.92 | 21.50 | 10.67 | 6.16 | 4.65 | 4.03 |
| GELU | 698.84 | 698.84 | 698.84 | 698.84 | 698.84 | 698.84 | 698.84 | 698.84 | 698.84 | 698.84 | 698.84 | 698.84 |
| Pruned GELU | 325.10 | 317.18 | 313.43 | 275.94 | 236.95 | 191.96 | 135.02 | 88.34 | 46.68 | 16.50 | 5.58 | 5.58 |

In Figure 19, we illustrate why CipherPrune can reduce the communication overhead of both SoftMax and GELU. Suppose there are $n$ tokens in $layer_i$. Then, the SoftMax protocol in the attention module has a complexity of $O(n^2)$. CipherPrune's token pruning protocol is invoked to select $n'$ tokens out of all $n$ tokens, where $m = n - n'$ is the number of tokens that are removed. The overhead

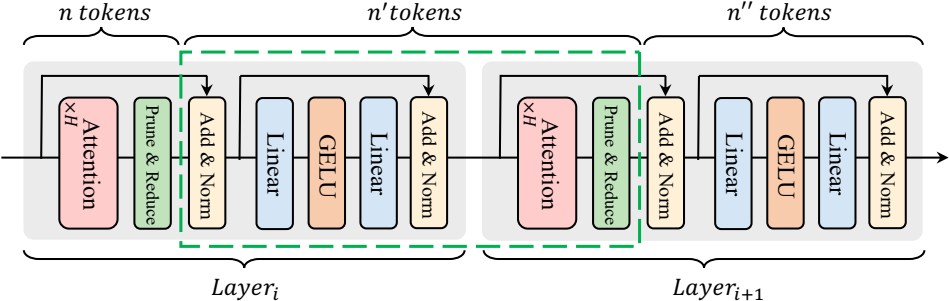

Figure 19: Toy example of two successive Transformer layers. In layer$_i$, the SoftMax and Prune protocol have $n$ input tokens. The number of input tokens is reduced to $n'$ for the Linear layers, LayerNorm and GELU in layer$_i$ and SoftMax in layer$_{i+1}$.

of the GELU function in $layer_i$, i.e., the current layer, has only $O(n')$ complexity (which should be $O(n)$ without token pruning). The complexity of the SoftMax function in $layer_{i+1}$, i.e., the following layer, is reduced to $O(n'^2)$ (which should be $O(n^2)$ without token pruning). The SoftMax protocol has quadratic complexity with respect to the token number and the GELU protocol has linear complexity. Therefore, CipherPrune can reduce the overhead of both the GELU protocol and the SoftMax protocols by reducing the number of tokens. In Table 3, we provide detailed layer-wise communication cost of the GELU and the SoftMax protocol. Compared to the unpruned baseline, CipherPrune can effectively reduce the overhead of the GELU and the SoftMax protocols layer by layer.

# F ANALYSIS ON LAYER-WISE REDUNDANCY.

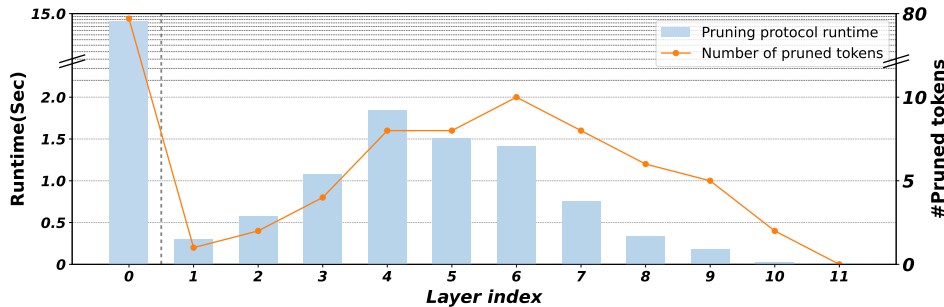

Figure 20: The number of pruned tokens and pruning protocol runtime in different layers in the BERT Base model. The results are averaged across 128 QNLI samples.

In Figure 20, we present the number of pruned tokens and the runtime of the pruning protocol for each layer in the BERT Base model. The number of pruned tokens per layer was averaged across 128 QNLI samples, while the pruning protocol runtime was measured over 10 independent runs. The mean token count for the QNLI samples is 48.5. During inference with BERT Base, input sequences with fewer tokens are padded to 128 tokens using padding tokens. Consistent with prior token pruning methods in plaintext (Goyal et al., 2020), a significant number of padding tokens are removed at layer 0. At layer 0, the number of pruned tokens is primarily influenced by the number of padding tokens rather than token-level redundancy.

In CipherPrune, tokens are removed progressively, and once removed, they are excluded from computations in subsequent layers. Consequently, token pruning in earlier layers affects computations in later layers, whereas token pruning in later layers does not impact earlier layers. As a result, even if layers 4 and 7 remove the same number of tokens, layer 7 processes fewer tokens overall, as illustrated in Figure 20. Specifically, 8 tokens are removed in both layer 4 and layer 7, but the runtime of the pruning protocol in layer 4 is $\sim 2.4\times$ longer than that in layer 7.

## G  RELATED WORKS

In response to the success of Transformers and the need to safeguard data privacy, various private Transformer Inferences (Chen et al., 2022; Zheng et al., 2023; Hao et al., 2022; Li et al., 2022; Lu et al., 2025; Luo et al., 2024; Pang et al., 2024) are proposed. To efficiently run private Transformer inferences, multiple cryptographic primitives are used in a popular hybrid HE/MPC method IRON (Hao et al., 2022), i.e., in a Transformer, HE and SS are used for linear layers, and SS and OT are adopted for nonlinear layers. IRON and BumbleBee (Lu et al., 2025) focus on optimizing linear general matrix multiplications; SecFormer Luo et al. (2024) improves the non-linear operations like the exponential function with polynomial approximation; BOLT (Pang et al., 2024) introduces the baby-step giant-step (BSGS) algorithm to reduce the number of HE rotations, proposes a word elimination (W.E.) technique, and uses polynomial approximation for non-linear operations, ultimately achieving state-of-the-art (SOTA) performance.

Other than above hybrid HE/MPC methods, there are also works exploring privacy-preserving Transformer inference using only HE (Zimerman et al., 2023; Zhang et al., 2024). The first HE-based private Transformer inference work (Zimerman et al., 2023) replaces SoftMax function with a scaled-ReLU function. Since the scaled-ReLU function can be approximated with low-degree polynomials more easily, it can be computed more efficiently using only HE operations. A range-loss term is needed during training to reduce the polynomial degree while maintaining high accuracy. A training-free HE-based private Transformer inference was proposed (Zhang et al., 2024), where non-linear operations are approximated by high-degree polynomials. The HE-based methods need frequent bootstrapping, especially when using high-degree polynomials, thus often incurring higher overhead than the hybrid HE/MPC methods in practice.

