# OpenReview forum: "CipherPrune:  Efficient and Scalable Private Transformer Inference"
_ICLR.cc/2025/Conference — ICLR 2025 Poster_

### Official Review · Reviewer_Xg5m · 2024-11-03

**Soundness:** 3
**Presentation:** 3
**Contribution:** 2
**Rating:** 5
**Confidence:** 2

**Summary:**

The paper introduces **CipherPrune**, a framework designed to enhance the efficiency and scalability of private inference in Transformer models, particularly BERT and GPT2, by implementing secure token pruning and polynomial reduction protocols. It addresses challenges related to runtime overhead and scalability by utilizing cryptographic techniques such as correlated oblivious transfer and secure multi-party computation, achieving significant speed improvements (up to 18.2x faster than existing methods) without compromising accuracy. Additionally, the framework incorporates crypto-aware fine-tuning to optimize pruning and approximation thresholds during training, ensuring efficient inference while maintaining data privacy.

**Strengths:**

- **Hybrid Cryptographic Approach**: The paper introduces a hybrid framework that combines homomorphic encryption for linear operations and secure multi-party computation (MPC) for non-linear operations, significantly improving the efficiency and scalability of private inference in Transformer models like BERT and GPT2.

- **Secure Token Pruning Protocol**: CipherPrune implements a secure token pruning protocol that evaluates token importance against a learned threshold, allowing for efficient pruning of unimportant tokens while preserving data privacy during inference.

- **Crypto-aware Fine-tuning**: The framework incorporates crypto-aware fine-tuning to optimize pruning and approximation thresholds during training, ensuring that the model maintains high performance and accuracy while facilitating efficient private inference.

**Weaknesses:**

I have a concern about the communication cost comparison between SoftMax and GELU here.

From the manuscript, the pruned tokens are selected after attention so that the communication from SoftMax here is still significant because you need to do compare, multiply, and etc.

How does the communication cost from SoftMax compared to one that GELU introduce?

**Questions:**

see weakness

---

> ### Author Response · Authors · 2024-11-23
> **Reply to Reviewer Xg5m**
>
> We thank Reviewer Xg5m for their careful review of our protocol and manuscript, as well as for their constructive comments.
>
> **Q1: The communication cost of GELU and SoftMax.**
>
> While it is true that the token pruning protocol is executed after the SoftMax protocol in each layer, we would like to clarify that CipherPrune significantly reduces the overhead of both the SoftMax and GELU protocols, as demonstrated in Figure 10 of the manuscript. This is because Transformers consist of multiple layers, each with its own SoftMax protocol. By pruning tokens, CipherPrune reduces the computational cost of the SoftMax protocols in subsequent layers. Additionally, since the GELU protocol always occurs after our token pruning protocol, the overhead of all GELU operations is reduced. The only exception is the SoftMax overhead in the first layer, which remains unchanged. It has a minimal effect on the proposed methods.
>
> To further highlight this improvement, we have included Figure 18 in Appendix E of the updated manuscript. This figure illustrates the reduction in overhead for these protocols.
>
> In addition to token pruning, CipherPrune also reduces the polynomial degree used in the GELU and SoftMax protocols through its polynomial reduction protocol, which is invoked after token pruning.
>
> We have conducted additional experiments to provide a detailed breakdown, presenting layer-by-layer communication costs for both the GELU and SoftMax protocols. These results clearly show the overhead reduction for pruned SoftMax (with our methods) compared to the original SoftMax, as well as for pruned GELU (with our methods) compared to the original GELU. The communication costs are presented in megabytes.
>
> | Layer | 1 | 2 | 3 | 4 | 5 | 6 |
> |-------|--------|--------|--------|--------|--------|--------|
> | Softmax (MB) | 642.19| 642.19 | 642.19 | 642.19 | 642.19 | 642.19 |
> | Pruned Softmax (MB) | 642.19 | 129.58 | 127.89 | 119.73 | 97.04 | 71.52 |
> | GELU (MB) | 698.84 | 698.84 | 698.84 | 698.84 | 698.84 | 698.84 |
> | Pruned GELU (MB) | 325.10 | 317.18 | 313.43 | 275.94 | 236.95 | 191.96 |
>
> | Layer | 7 | 8 | 9 | 10 | 11 | 12 |
> |-------|--------|--------|--------|--------|--------|--------|
> | Softmax  (MB)| 642.19 | 642.19 | 642.19 | 642.19 | 642.19 | 642.19 |
> | Pruned Softmax  (MB) | 43.92 | 21.50 | 10.67 | 6.16 | 4.65 | 4.03 |
> | GELU (MB) | 698.84 | 698.84 | 698.84 | 698.84 | 698.84 | 698.84 |
> | Pruned GELU (MB) | 135.02 | 88.34 | 46.68 | 16.50 | 5.58 | 5.58 |

---

> > ### Author Response · Authors · 2024-12-02
> > **We are keen for your response**
> >
> > Dear Reviewer Xg5m,
> >
> > We genuinely thank you again for your time, efforts, and constructive comments. We appreciate your overall positive comments in your initial review.
> >
> > In our response, with additional data, clarification, and illustration, we have addressed your concern about the communication cost of SoftMax and GELU.
> >
> > As the discussion period is approaching its end, we would really appreciate it if you could kindly let us know whether there are any further questions. We will be more than happy to address them.
> >
> > Best wishes,
> >
> > The Authors

---

> ### Author Response · Authors · 2024-11-26
> **Rebuttal follow-up**
>
> Dear Reviewer Xg5m,
>
> Thank you for your comments on our paper. We hope the additional data and clarifications we provided have adequately addressed the concerns raised in your review. We would greatly appreciate your feedback on our responses.
>
> Best regards,
>
> The Authors

---

### Official Review · Reviewer_Mztk · 2024-11-04

**Soundness:** 3
**Presentation:** 3
**Contribution:** 3
**Rating:** 6
**Confidence:** 4

**Summary:**

This paper introduces an innovative framework designed to address the efficiency and scalability challenges of private Transformer inference. The proposed CipherPrune framework incorporates secure encrypted token pruning and polynomial reduction protocols to optimize performance. By progressively pruning unimportant tokens and assigning lower-degree polynomial approximations to less important tokens, CipherPrune achieves notable reductions in execution time and communication costs without compromising accuracy.

**Strengths:**

1. The combination of encrypted token pruning and polynomial reduction is novel and well-executed, contributing to reduce PI costs and latency while maintaining model performance.

2. The proposed CipherPrune demonstrates substantial improvements in runtime and communication efficiency.

**Weaknesses:**

As token pruning may be less effective in tasks that rely heavily on retaining comprehensive token-level information, such as machine translation, I suggest that the authors explicitly outline which applications are most suitable for this method and where its benefits may be more limited. This clarification would enhance the practical understanding and applicability of the work.

**Questions:**

Can you specify which types of NLP tasks or applications are most suitable for the CipherPrune framework, and which tasks may see limited benefits from the proposed token pruning approach?

---

> ### Author Response · Authors · 2024-11-23
> **Reply to Reviewer Mztk**
>
> We thank Reviewer Mztk for the thorough reading of the manuscript and constructive comments.
>
> **Q1: Discuss tasks that are most suitable for CipherPrune.**
>
>
>
> For token pruning, it generalizes effectively to both discriminative and generative models and tasks (e.g., language modeling and machine translation). While applying token pruning to discriminative tasks is relatively straightforward, its application to generative models and tasks might initially seem counterintuitive. However, there is a key reason why token pruning is beneficial for generative tasks: each token generation in a generative task operates similarly to a discriminative task.
>
> In a discriminative task using a BERT-based Transformer encoder, the model produces a single summarization for classification. Conversely, in a generative task, the model relies on an optional encoder summarization and performs multiple-iteration generation. Each iteration generates one token, taking the summarization and previously generated tokens as inputs. The initial summarization typically requires comprehensive token information. However, in later generation iterations, many tokens can be pruned significantly, as not all tokens contribute equally to generating each subsequent token. This property allows intermediate tokens to be pruned significantly during generation tasks.
>
> As demonstrated in prior works [1, 4, 5, 6, 7], token pruning is effective for generative models and tasks, often producing results comparable to discriminative models [1, 2, 3]. Our experiments further confirm this, showing that CipherPrune reduces the execution overhead for private GPT-2 language modeling tasks by approximately 5.3× for 128-token inputs and 8.2× for 512-token inputs. We will include this result and explore more models and tasks in the next version of the manuscript.
>
>
> [1] Wang et al., Spatten: Efficient sparse attention architecture with cascade token and head pruning.
>
> [2] Kim et al,  Learned token pruning for transformers
>
> [3] He et al, Magic pyramid: Accelerating inference with early exiting and token pruning.
>
> [4] Anagnostidis et al.,  Dynamic context pruning for efficient and interpretable autoregressive transformers.
>
> [5] Zhang et al., H2o: Heavy-hitter oracle for efficient generative inference of large language models.
>
> [6] Oren et al., Transformers are multi-state rnns.
>
> [7] Fu et al., Lazyllm: Dynamic token pruning for efficient long context llm inference.

---

> > ### Comment · Reviewer_Mztk · 2024-11-29
> > **Thanks for your response**
> >
> > I would like to thank the authors for the response. I will maintain the scores as they currently stand.

---

### Official Review · Reviewer_V3E7 · 2024-11-06

**Soundness:** 4
**Presentation:** 4
**Contribution:** 3
**Rating:** 6
**Confidence:** 4

**Summary:**

This paper introduces *CipherPrune*, a framework designed to mitigate the communication and latency overheads associated with both nonlinear and linear operations (FLOPs) in private inference for transformer-based models. CipherPrune introduces a token pruning method that assesses input token redundancy, progressively removing unimportant tokens layer by layer across the network, thereby leveraging task-level token redundancy to achieve efficiency without sacrificing predictive performance.

Additionally, for less important tokens CipherPrune employs low-degree polynomial approximation for GELU and Softmax, in contrast with the higher degree polynomial approximation for critical input tokens, further enhancing efficiency without compromising security guarantees or predictive performance.

**Strengths:**

$\bullet$ This work contributed to both the front-- protocol and network level-- and demonstrated the potential of a synergistic approach that achieves significant speedups and substantial communication reductions.

$\bullet$ The authors identify limitations in plaintext-based token pruning for private inference and introduce ciphertext-specific pruning techniques that avoid the need for decryption at each network layer, addressing a key bottleneck in efficiency.

$\bullet$ They effectively tackle the shortcomings of task-agnostic token pruning (such as word elimination) seen in prior work, demonstrating the advantages of a task-specific, layer-wise progressive token pruning strategy that optimizes both performance and efficiency.

$\bullet$ The paper is well-written and clearly outlines the challenges, the limitations of previous approaches, and the authors' contributions, making it easy to follow and appreciate the impact of this work.

**Weaknesses:**

$\bullet$ Step 3 (Figure 4), prune and reduce, would increase the computational burden on the client side. A discussion on the resource burden (computation and memory) on the client should have been included, assuming the client has a low-end mobile device with limited compute and memory.

$\bullet$ This work does not address the overhead introduced by LayerNorm, which can add substantial communication and latency costs depending on the protocols used. For instance, in CipherGPT [3], LayerNorm accounts for **22%** of the total latency and communication overhead.

$\bullet$ One significant drawback of this paper is the absence of a detailed characterization of layerwise token redundancy. Without this, readers miss out on key insights into how redundancy varies across layers, which could have helped guide more efficient network design and optimization.

$\bullet$ Overall, the CipherPrune framework appears quite complex to understand, implement, and deploy. Its successful application requires expertise in both protocol design and network architecture, making it challenging for those without specialized knowledge in these areas.



## Correction in the draft

$\bullet$ L#355 Pruning ratio larger than zero --> Pruning ratio greater than zero

$\bullet$ Figure 4 depicts the 2PC threat model for Post-LN configuration, and in the FFN block diagram, one linear layer (after the GELU layer) is missing.

$\bullet$ Missing citations [1] and [2]

1.  Zimerman et al., Converting Transformers to Polynomial Form for Secure Inference Over Homomorphic Encryption,  ICML 2024

2.  Luo et al. SecFormer: Towards Fast and Accurate Privacy-Preserving Inference for Large Language Models, ACL 2024.

3. Hou et al., CipherGPT: Secure Two-Party GPT Inference

**Questions:**

See the weaknesses

---

> ### Author Response · Authors · 2024-11-23
> **Reply to Reviewer V3E7**
>
> We thank Reviewer V3E7 for the thorough reading of the manuscript and for providing constructive comments.
>
> **Q1: Discussion of client-side overhead for Step 3 of Figure 4.**
>
> The introduced pruning imposes only marginal overhead on the client side. In Figure 10 of our manuscript, we provide a detailed breakdown of the execution time for each computational component, including the introduced pruning. The figure demonstrates that the pruning overhead accounts for less than $5\\%$ of the total runtime, and this overhead ratio is consistent for the client-side as well. Additionally, we have included specific client-side overhead data for pruning and reduction in our experiments, which are based on the BERT Base model in a WAN setting.
>
> |                 | Time | Peak Memory | Communication |
> |-----------------|------|--------|---------------|
> | Score Computing | 2.33ms | 4.5MB       | 0   |
> | Prune | 571.99ms | 11.27MB             | 3.13MB |
> | Reduction | 10.1ms | 10.77MB           | 36.15KB |
> | Total | 584.53ms | 11.27MB             | 3.16MB |
>
>
>
> **Q2: The overhead of LayerNorm may not be reduced**
>
> We clarify that CipherPrune effectively reduces the overhead of LayerNorm, as the number of tokens is decreased through encrypted token pruning. Since the complexity of LayerNorm is linear with respect to the number of tokens, this reduction directly lowers its computational overhead. Figure 10 in our manuscript illustrates this reduction in LayerNorm overhead.
>
> **Q3: Add characterization of layerwise token redundancy.**
>
> We appreciate the reviewer’s suggestion to include this characterization. In Figure 19 in Appendix F, we present the detailed number of tokens at each layer from our experiments, along with the runtime of the secure pruning protocol. Our findings indicate that intermediate layers tend to exhibit more redundancy, leading to a higher number of tokens being pruned at these stages. We believe this data provides valuable insights into layer-wise token redundancy and can inform future research in this area.
>
> **Q4: The deployment difficulties due to protocol-network codesign.**
>
> Deploying CipherPrune in practice is straightforward. CipherPrune leverages modular block operations built on fundamental cryptographic primitives, such as oblivious transfer-based comparison and MSB extraction. We already offered the open-source code to enhance reproducibility and facilitate deployment.  We plan to elaborate the readme and add a tutorial.
>
>
> **Editorial improvements and citations.**
>
> * We have corrected the typo at line 355 within the manuscript, highlighted in blue.
> * We use the same configuration with previous works such as Bolt for the Post-LN. We have updated the FFN module in Figure 4 for better clarity.
> * We thank Reviewer V3E7 for pointing out the missing citations. We have added these citations and discussed them in the related work section for a more comprehensive literature review.

---

> ### Comment · Reviewer_V3E7 · 2024-11-24
> **Official Comments for Authors**
>
> Thank you for providing the rebuttal.
>
> >Q2: The overhead of LayerNorm may not be reduced
>
> By pruning input tokens, you have reduced the number of LayerNorm operations, which is different from making the operations themselves faster, hence there is a distinction. By pruning tokens, both linear and nonlinear operations have been reduced, which is different from making the operations themselves more efficient.
>
> I have one  question on the newly added results (Figure 19)
>
> >This is because there are more input tokens in the early layers, resulting in more secure swap operations. For example, 8 tokens are removed in both layer 4 and layer 7, but the protocol runtime in layer 4 is ∼ 2.4× longer than that in layer 7.
>
> How does removing the same number of tokens from two layers lead to different input token counts in those layers? Also, why there are only 11 layers in the BERT-base model?
>
> **Minor**
> Please correct the BibTeX entry for the BumbleBee paper (It's accepted to NDSS 2025)

---

> > ### Comment · Reviewer_V3E7 · 2024-11-25
> > **Thanks for clarification**
> >
> > Thanks for the clarification on the layer 0 redundancy characteristics. Prior work on Mechanistic Interpreabality also found that the first layer (layer 0) in LLM has a distinct characteristic and their neurons are harder to interpret.
> >
> > The authors have explained it very well, and I would suggest the authors to include these discussions in the final version.
> >
> > Given the complexity and limited novelty of the proposed method, I will maintain my score.

---

> > > ### Author Response · Authors · 2024-11-25
> > > **Reply to Reviewer V3E7**
> > >
> > > Thank you for confirming that we have addressed the questions effectively. We truly appreciated the discussion and are grateful for the insightful suggestions.
> > >
> > > Privacy-preserving machine learning based on cryptography is an interdisciplinary yet crucial area of research, requiring significant expertise to advance and substantial effort to develop methods and open-source codes (We have accomplished this with significant dedication and effort). A more positive score could encourage greater engagement from students and researchers, fostering progress in this important field.

---

> ### Author Response · Authors · 2024-11-25
> **Follow-up Reply to Reviewer V3E7**
>
> We appreciate the reviewer’s follow-up comments.
>
> **Overhead of LayerNorm protocol.**
>
> That is correct. The token pruning reduces the overhead of the total LayerNorm operations in the network inference, but does not reduce each LayerNorm's overhead.  A more efficient LayerNorm protocol will further enhance the efficiency. We will clarify this more explicitly in our final version.
>
> **Number of tokens in Figure 19: How does removing the same number of tokens from two layers lead to different input token counts in those layers? 11 Layers (BERT) in Figure 19.**
>
> In CipherPrune, tokens are removed progressively, and once removed, they are excluded from computations in subsequent layers. Consequently, token pruning in earlier layers affects computations in later layers, whereas token pruning in later layers does not impact earlier layers. As a result, even if layers 4 and 7 remove the same number of tokens, layer 7 processes fewer tokens overall, as illustrated in Figure 19. BERT consists of 12 layers (layer 0 to layer 11). In the original Figure 19, we focused on layers 1 through 11 because layer 0 is distinct in its redundancy characteristics. Specifically, redundancy in layer 0 arises from both padding tokens (when the input length is shorter than the maximum sequence length, typically 128 in prior works like BOLT [1]) and the input itself. In contrast, redundancy in layers 1–11 is primarily derived from the input alone, as padding tokens are already removed in layer 0. Including only layers 1 through 11 allowed us to emphasize redundancy patterns specific to the input. However, to address concerns and improve clarity, we have now added layer 0 to the updated Figure 19 and its description, ensuring a more comprehensive representation of pruning behavior across all layers.
>
> We are happy to provide further clarification or follow-up responses if needed.
>
> [1] Pang et al., BOLT: Privacy-Preserving, Accurate, and Efficient Inference for Transformers.

---

### Official Review · Reviewer_Qed2 · 2024-11-08

**Soundness:** 4
**Presentation:** 4
**Contribution:** 4
**Rating:** 8
**Confidence:** 4

**Summary:**

CipherPrune effectively removes unnecessary tokens by combining input-adaptive pruning based on importance scores with progressive layer-wise pruning. For the remaining tokens, it applies high-order and low-order polynomials according to their importance to maximize computational efficiency. This approach achieved 6.1x and 10.6x faster performance on inputs of 128 and 512 tokens, respectively, significantly improving inference speed without accuracy loss.

**Strengths:**

- Achieving a substantial improvement in speed, with a 6.1x increase for 128 tokens and a 10.6x increase for 512 tokens, without any reduction in accuracy, is remarkable.

- The application of different polynomial orders based on importance scores, rather than merely pruning, is an impressive idea.

- The paper clearly differentiates itself from existing studies by detailing each step, explaining what previous research overlooked, why these aspects are necessary, and how their work contributes to these areas.

- All experimental settings are well-documented, providing extensive end-to-end results and comparisons across multiple models.

This paper demonstrates advancements over BOLT in every aspect. The time complexity of pruning and the resulting speed have both improved, with no difference in accuracy compared to BOLT. The authors provided a thorough analysis of BOLT, detailing the shortcomings of its ideas and the specific areas they have enhanced. In addition to building on ideas from previous studies, they introduced an innovative approach by applying polynomials of different orders based on importance scores. Comparisons with various papers are well-presented in the appendix, and it’s impressive that this technique can be applied to most inference models, including 2PC and 3PC models. I believe this is an excellent paper.

**Weaknesses:**

- Although the paper claims no accuracy reduction, there is a minor decrease of around 0.2%.

- In terms of pruning, the only prior work compared is BOLT. (I’m unsure if there are other relevant papers, so it may be possible that only BOLT exists in this context.)

**Questions:**

See the weakness

---

> ### Author Response · Authors · 2024-11-23
> **Reply to Reviewer Qed2**
>
> We thank Reviewer Qed2 for the thorough reading of our manuscript and for providing constructive comments.
>
> **Q1: The 0.2\% accuracy decrease.**
>
> We appreciate the reviewer pointing out that our methods may result in a marginal accuracy drop, rather than no accuracy drop as initially claimed. We have revised the manuscript to make this claim more precise and accurate.
>
> **Q2: Prior works on private token pruning.**
>
> Pruning on encrypted data remains a relatively new area for the research community. At the time of submission, BOLT [1] was the only relevant work we could identify. We have clarified the distinctions between our work and BOLT, and we highlight the improvements our method offers over BOLT.
>
> We believe that CipherPrune represents an innovative contribution to the field, demonstrating the potential of crypto-machine learning co-design and inspiring future research in this area.
>
> [1] Pang et al., BOLT: Privacy-Preserving, Accurate, and Efficient Inference for Transformers.

---

### Meta-Review · Area_Chair_Z1kS · 2024-12-23

**Metareview:**

All reviewers except one (Xg5m) argued for accepting the paper. For this reviewer their main conerns were on the communication cost comparison between SoftMax and GELU. The authors clarified that the pruning protocol is able to reduce the overhead of both the SoftMax and GELU protocols. They added an additional Figure 18 to highlight this. The reviewer did not reply further, and I find that this concern is dealt with. The paper shows impressive speed-up results, is clearly written, and introduces an interesting new method. Given these things, I vote to accept. Authors: you’ve already made improvements to respond to reviewer changes, if you could double check their comments for any recommendation you may have missed on accident that would be great! The paper will make a nice contribution to the conference!

**Additional Comments On Reviewer Discussion:**

Two reviewers responded to the author feedback (V3E7, with further clarification questions and Mztk, to say they will maintain their score). No reviewers engaged in further discussion of the paper. For more detail on how I made the decision please see the metareview. Reviewer Mztk's review was so short I disregarded it. I would not recommend inviting them to a future review cycle.

---

### Decision · Program_Chairs · 2025-01-22

Accept (Poster)